# The Challenges for EU User Testing Policies for Patient Information Leaflets

**DOI:** 10.3390/ijerph21101301

**Published:** 2024-09-29

**Authors:** Nicola Pelizzari

**Affiliations:** School of Education, Languages and Linguistics, Faculty of Humanities and Social Sciences, University of Portsmouth, Park Building, King Henry I St., Portsmouth PO1 2BZ, UK; up2090530@myport.ac.uk

**Keywords:** patient information leaflets, user testing, healthcare communication, EU regulations, translation policies

## Abstract

Patient information leaflets (PILs) are essential tools in healthcare, providing crucial information about medication use. In the European Union, the European Medicines Agency (EMA) oversees the regulation and standardisation of PILs to ensure their readability and accessibility. However, challenges persist in ensuring these documents are comprehensible and user-friendly. This study employs a qualitative analytical approach, reviewing existing literature and regulatory documents to identify gaps in the EU user testing policies for PILs. It focuses on the diversity of participant samples, the independence of the testing process, and the robustness of user testing protocols. Findings indicate that current user testing practices often lack diversity and may be biased when pharmaceutical companies conduct their own tests. Additionally, there is a lack of user testing protocols for translated PILs, potentially compromising their accuracy and cultural relevance. To improve the efficacy of PILs, it is essential to include diverse and representative samples in user testing, mandate independent third-party evaluations, implement protocols for user testing on translated PILs, and ensure continuous updates to guidelines based on the latest best practices in health communication. These measures will enhance patient safety and understanding of medication information.

## 1. Introduction

Patient information leaflets (PILs), also known as package leaflets, are critical tools in healthcare communication, designed to ensure that patients can safely and effectively use their medications. These documents included in medical packaging provide essential information, including dosage instructions, potential side effects, and storage conditions [1]. The creation and regulation of PILs are influenced by a complex interplay of institutional and contextual factors, aiming to standardise and enhance their readability and accessibility [2]. In Europe, the regulation of these documents is primarily overseen by the European Medicines Agency (EMA), which is responsible for the evaluation and supervision of medicines to ensure their safety, efficacy, and quality [3]. The EMA collaborates with regulatory authorities from the European Economic Area (EEA) countries, the European Commission, and other stakeholders. This collaborative network ensures a consistent approach to the approval and monitoring of medicinal products across Europe [4]. The EMA’s efforts are complemented by various national regulatory bodies that implement these standards within their respective countries [5]. The concept of PILs, as we know them today, originated in the late 1960s when regulatory bodies recognised the need for clear communication regarding medication use. In 1968, the Food and Drug Administration (FDA) in the United States required an inhalation drug to include a warning about excessive use [6]. This marked the beginning of a broader movement towards enhancing patient information. By the 1970s, the FDA mandated that contraceptive pills include detailed information on risks and benefits [7]. This trend was mirrored in Europe, where PILs began to appear for over-the-counter drugs and medications used without direct medical supervision. The regulation of PILs in Europe has undergone significant evolution. Initially, each country had its own legislative framework for PILs. However, European Union reforms introduced standardised guidelines to harmonise these regulations [8]. The mandatory nature of PILs was established with Council Directive 92/27/EEC in 1992, which became legally binding in 1999 [5]. These guidelines stipulated that PILs must include comprehensive information about the medication, such as drug identification, therapeutic indications, precautionary measures, dosage and administration methods, side effects, storage conditions, and expiry dates. Another document, the Summary of Product Characteristics (SmPC) serves as a core document, providing detailed information for healthcare professionals and ensuring the safe and effective use of medications [9]. One of the key requirements for PILs, as mandated by Article 63(2) of Directive 2001/83/EC, is that they must be written in clear and understandable terms for the general public [10]. This directive emphasises the need for PILs to be easily comprehensible and legible in the official language or languages of the member state where the medicinal product is marketed. This regulation aims to eliminate barriers to understanding and ensure that patients can access critical information in a language they can easily comprehend [11]. PILs are categorised as compulsory genres; namely, documents which are subject to stringent legal regulations and legislative mandates [12]. In the European context, the sale of pharmaceutical products is contingent not only on the verification of their efficacy and safety but also on the accompanying informational materials, including the outer packaging, labels, and PILs [13]. The authorisation to sell a medicine is granted by the relevant regulatory agency, which ensures that all requirements are met [14]. The marketing authorisation process in the European Union involves a network of regulatory authorities collaborating to evaluate new medicines and share safety information [14]. In the European Union, pharmaceutical companies can pursue different procedures for obtaining marketing authorisation. The centralised procedure, overseen by the EMA, allows companies to submit a single application for EU-wide approval, mandatory for certain medicines, including biotechnology products and orphan drugs. Alternatively, the decentralised procedure enables companies to apply for authorisation in multiple member states simultaneously, with one reference member state (RMS) leading the assessment, and other concerned member states (CMSs) either accepting or rejecting the decision. This route is typically used for medicines not previously authorised in any EU country and outside the centralised procedure’s scope. The mutual recognition procedure, by contrast, is applied when a product has already been authorised in one EU member state, allowing the company to extend the marketing authorisation to other member states, which then “mutually recognise” the existing authorisation. Each option offers a tailored approach depending on the product type and market strategy [15].

In whichever route is chosen to apply for authorisation, whether through the centralised, decentralised, or mutual recognition procedure, the EMA ultimately conducts a scientific evaluation and provides its opinion to the European Commission for the final decision. Once approved by the European Commission, the centralised marketing authorisation is valid in all EU member states [15]. The introduction of PILs in 1992 triggered a continuous improvement process from the patient’s perspective. Regulatory bodies have become increasingly aware of the importance of clear language in PILs and the challenges associated with producing easy-to-understand documents, such as addressing varying levels of health literacy among patients, ensuring the accuracy and completeness of information, and balancing the need for comprehensiveness with the necessity of maintaining readability [16]. Various initiatives have been implemented to enhance the readability and accessibility of PILs. For instance, guidelines have been established regarding type size and font, design and layout, headings, print colour, syntax, and style [9]. These guidelines aim to ensure that PILs are accessible to a broad audience, including those with poor reading skills and some degree of vision loss [9]. One of the significant improvements in PILs has been the implementation of user testing. This process involves testing the readability of a specimen with a group of selected test subjects to identify whether the information presented is easily understandable [17]. User testing does not improve the quality of the information itself but indicates areas where modifications are necessary [18]. For a PIL to be approved, it must pass the user test, demonstrating that at least 90% of participants can find and understand the information correctly. Despite these advancements, challenges remain in producing satisfactory PILs. Studies have shown that even well-designed and clearly worded PILs can be difficult for some users to understand [16,19,20,21]. Issues often arise from the focus on content rather than language, highlighting the need for ongoing improvements [22]. The European Union’s mandatory user test for PILs, introduced in 2005, aims to address these concerns by ensuring that PILs are tested with a sample of users to verify their readability [8,23]. However, the criteria for evaluating these tests can be vague, leaving room for interpretation and variation in testing methods [24]. The aim of this study is to critically examine the current EU policies surrounding user testing for PILs and identify gaps that may hinder their effectiveness in ensuring readability and accessibility for diverse patient populations. By evaluating the diversity of user testing samples, the independence of the testing process, and the sufficiency of testing protocols, this research seeks to offer recommendations for improving the robustness of these regulations to enhance patient safety and comprehension.

## 2. Methods

### 2.1. Study Design

This study employs a qualitative analytical approach to systematically review and analyse the current user testing policies for PILs within the European Union. The analysis focuses on identifying potential gaps and weaknesses in the regulatory framework as of June 2024. This approach is particularly suited to exploring the intricacies of policy development and implementation, as it allows for a comprehensive examination of both the content and context of the existing regulations. The legal framework for user testing in the EU is primarily established through several key legislative documents. This study focuses on Directive 2001/83/EC and its subsequent amendments (Revisions 14 through 14.8), which provide guidance on consultations with target patient groups for the package leaflet, and Articles 59(3) and 61(1) of Directive 2001/83/EC as amended by Directive 2004/27/EC, which outline the requirements for ensuring that PILs are legible, clear, and easy to use. These documents were selected based on their relevance to the user testing process and their role in shaping the current regulatory landscape for PILs.

### 2.2. Theoretical Framework 

Following the qualitative analytical framework outlined by Creswell [25], the analysis proceeded through several stages. First, relevant legislative documents were identified and collected through a comprehensive search of the European Union’s legal databases and related repositories, including both the primary directives and any relevant amendments or guidance documents. Each document was then subjected to a thorough content analysis, where key themes, concepts, and regulatory requirements were identified. 

### 2.3. Qualitative Data Analysis

Findings were recorded manually and subsequently compared against best practices in health communication and user testing, as established in the literature [26,27,28,29]. This comparative analysis allowed for the identification of gaps and inconsistencies in the current EU regulations. The results of the analysis were synthesized descriptively, focusing on key points that could impede the effectiveness of the user testing process. This synthesis aimed to develop a comprehensive understanding of the regulatory landscape and provide evidence-based recommendations for improving the user testing of PILs. To ensure the reliability and validity of the analysis, several strategies were employed. Multiple sources of data, including legislative texts, scholarly articles, and industry reports, were used to cross-check and corroborate findings. The researcher maintained a reflexive stance throughout the analysis, critically reflecting on how personal biases and assumptions might influence the interpretation of the data.

## 3. Findings

### 3.1. User Testing Origins and EU Adoption

The concept of user testing for PILs originated in the early 1990s in Australia [30]. By involving actual users in the testing process, developers could identify areas of confusion and make necessary adjustments, thus improving the overall quality of the information. This method proved successful and was subsequently incorporated into European guidelines, leading to its mandatory adoption within the European Union. Recognizing the effectiveness of this approach, the EU incorporated it into their regulatory framework through Directive 2001/83/EC (Appendix A), which was later amended by Directive 2004/27/EC. These directives mandate that all medicinal products must be accompanied by a PIL that reflects the results of consultations with target patient groups to ensure its legibility and ease of use. 


*“Articles 59(3) and 61(1) of Directive 2001/83 require that the patient information leaflet shall reflect the results of consultations with target patient groups to ensure that it is legible, clear and easy to use and that these results of assessments carried out in cooperation with target patient groups shall also be provided to the competent authority. They do not define the precise method to be used. As a consequence, these provisions permit user testing as well as other appropriate forms of consultation”.*


The European Medicines Agency oversees the implementation of these regulations, providing detailed guidelines on the preparation and testing of PILs. 

### 3.2. User Testing Process and Criteria

The process of user testing for PILs involves several critical steps (See Figure 1) and the participation of various stakeholders. 

According to the EMA’s Guideline on the Readability of the Label and Package Leaflet of Medicinal Products for Human Use (Appendix A), the main objective of these directions is to assist pharmaceutical companies in creating PILs that are accessible and comprehensible to all patients, including those with limited literacy skills or visual impairments. The user testing of PILs is a mandatory requirement for all new marketing authorisations granted after 30 October 2005. The process is detailed in Chapter 3 of the readability guideline, which specifies that the PIL must reflect the results of consultations with target patient groups. This user consultation can take various forms, with user testing being one of the most common and recommended methods. User testing involves testing the readability of a PIL specimen with a group of selected participants who represent the target patient population. The testing procedure is typically carried out by the marketing authorisation holder or by an external company contracted for this purpose. 

“*User testing of PILs must be organised by the marketing authorisation holders. (…) The responsibility for ensuring the readability and comprehensibility of PILs falls on the marketing authorisation holders*”.

The guideline suggests conducting one-to-one, face-to-face interviews with at least 20 participants. The participants should be representative of the patient population, excluding those directly involved with medicines such as healthcare professionals. The guideline emphasises the importance of using a full-colour mock-up of the PIL as it will be supplied with the marketed product to ensure similar testing conditions. During the testing process, participants are asked to locate specific pieces of information within the PIL and explain their understanding of it. The interviewer records their responses and observes their interaction with the leaflet, noting any difficulties in navigation or comprehension. The success criteria for the test require that 90% of participants can find the information and that 90% of those can understand and act appropriately based on that information. This means that at least 16 out of 20 participants must successfully complete each task for the PIL to be considered readable and user-friendly. 

### 3.3. Reporting and Implementation Challenges

Once the testing is complete, the results are compiled into a report, which is included in the application dossier submitted to the competent authority. 

“*The presentation of results should be shortened to a summary explaining how the consultation was executed and how the resulting patient information leaflet accommodated any need for change*”.

The report must summarise how the consultation was executed, detail the methodology used, and include the questionnaire and observations. Results of such consultation should be presented in English for the centralised, decentralised, and mutual recognition procedure, or in the national language for national procedures, to permit the assessment of the test to be undertaken by competent authority responsible for granting the marketing authorisation. In the centralised, decentralised, and mutual recognition procedure, only the English language version of the PIL will be agreed upon during the scientific assessment. Once the PIL in the original language is approved, user testing is not necessary for all subsequent translated versions.

“*As a matter of principle, it is sufficient to undertake patient consultation in one EEA language. (…) Following the grant of the marketing authorisation, the responsibility for the production of faithful translations will rest with the marketing authorisation holder in consultation with the Member States/European Medicines Agency*”.

The EMA or the national competent authority evaluates the user testing report as part of the overall assessment of the marketing authorisation application. The authorities check for evidence that the PIL is understandable and useful to patients and may request further modifications if the test results do not meet the required standards. While these directives and guidelines lay a robust foundation for the creation and assessment of PILs, the journey from regulatory text to practical implementation is riddled with complexities and challenges. Beneath the surface of this well-structured regulatory framework lies a labyrinth of potential pitfalls and hurdles that can undermine the ultimate goal of clear, user-friendly patient information. It is within this intricate dance between compliance and real-world application that several issues begin to emerge, revealing cracks in the seemingly solid edifice of the EMA’s regulatory oversight. These challenges, if left unaddressed, threaten to compromise the effectiveness and reliability of PILs, leaving patients with more questions than answers.

## 4. Discussion

### 4.1. Participant Selection and Diversity in User Testing

As the findings highlight, despite the invaluable benefits of the user testing process, current guidelines present several issues and gaps that may affect the final reliability and effectiveness. One significant concern is the pharmaceutical company’s prerogative to choose the composition of focus group for testing. The EMA guidelines only provide generic instructions, open to interpretability on how to select participants, which can result in non-representative samples. This absence of specificity means that the diversity of patient populations—including variations in age, literacy levels, cultural backgrounds, and specific medical conditions—may not be adequately captured, potentially compromising the generalizability of the findings. Demiris and Hensel emphasise that capturing this diversity is crucial for ensuring the reliability of the results [31]. Moreover, previous literature has demonstrated that non-representative user samples can lead to significant usability issues, especially in patient information leaflets. For example, studies by Albassam and Hughes as well as Liu et al. have identified gaps in PIL comprehension related to literacy levels and cultural background, where non-diverse samples led to poor real-world usability, ultimately compromising patient safety [19,24].

### 4.2. Potential Biases in the User Testing Process

Potential biases in the user testing process present another significant challenge. When pharmaceutical companies conduct their own user testing, there is a potential for conflict of interest, which may influence the objectivity of the process. This setup can lead to less rigorous testing and biased reporting [32], as the companies have a vested interest in seeing their products pass with flying colours. Baines [33] suggests that independent third-party testing can serve as an effective antidote to this bias, providing a layer of objectivity that internal company testing lacks. However, as this practice is seldom mandated [34], the lack of strict requirements for third-party involvement, the temptation for companies to take shortcuts or present overly optimistic results may remain high. In fact, Lexchin et al. found that industry-sponsored research tends to report more favourable outcomes compared to independent studies, potentially leading to overestimated usability and effectiveness of PILs. This underscores the critical need for independent testing to mitigate biased reporting [35].

Critically, the very structure of internal testing can be subtly, yet significantly, skewed. For instance, companies might—either intentionally or inadvertently—select test participants who are more likely to provide favourable feedback undermining the generalizability of the test results. Such biases have been observed in other pharmaceutical contexts, where selective publication of positive results is more common, further highlighting the risks of relying solely on industry-conducted tests [36]. Moreover, the pressures of commercial success can lead to optimistic interpretations of data in line with studies by Lexchin et al. [35] which found that industry-sponsored research is more likely to report positive findings compared to independent studies. Alonso also notes how conflicts of interest in pharmaceutical evaluations can result in misleading claims, exacerbating usability problems in critical health materials such as PILs [37]. This is not to say that pharmaceutical companies are unscrupulous, but when profits are at stake, the line between rigorous science and convenient conclusions can blur [37,38,39].

Independent third-party testing should be the norm rather than the exception. Such testing could involve multiple stakeholders, including patient advocacy groups, regulatory bodies, and independent research organisations, bringing a level of scrutiny and impartiality that internal testing simply cannot match, consequently building public trust and bolstering the credibility of the findings. 

### 4.3. The Controlled Environment in User Testing

Another issue related to regulations is the environment in which user testing occurs, potentially influencing the results, as it may not accurately reflect real-world conditions where patients interact with PILs. Ideally, user testing should replicate real-world scenarios as closely as possible to capture authentic patient experiences and behaviours [26]. 

However, achieving this in a controlled testing environment poses considerable challenges. Controlled environments are often designed to eliminate variables that could affect the outcome of the tests, providing a consistent and repeatable setting for evaluation. While this consistency is beneficial for isolating the impact of specific variables, it can also create an artificial context that does not mirror the complexities of everyday life [40]. According to Tromp et al. [41], usability testing in artificial settings may fail to account for the diverse contexts in which users engage with products, leading to results that do not fully represent real-world usability. For example, in real-life situations, patients may read PILs under various conditions, such as when they are ill, at home with distractions, or while experiencing stress or anxiety related to their health. These factors can significantly influence how patients comprehend and retain the information. Andargoli et al. [42] highlighted that environmental context plays a crucial role in the usability of health information systems, suggesting that testing should incorporate elements that mimic real-life conditions to better understand user interactions and challenges. Additionally, the presence of testers and the structured nature of the testing process can impact how participants respond, often due to the Hawthorne effect, where individuals alter their behaviour because they are aware of being observed [43]. This phenomenon, as described by McCarney et al. [44], can lead to changes in performance or productivity, potentially skewing research outcomes. This effect should be taken into consideration as it can lead to overestimation of the comprehensibility and usability of PILs, as participants might pay more attention or exert more effort than they would in their everyday lives. To mitigate these issues, it is essential to incorporate real-world elements into the testing process. One approach may be remote user testing [45], a method allowing participants to engage with the materials in their own settings, which can lead to more authentic feedback and a better understanding of real-world usability challenges. Studies have demonstrated that remote usability testing can effectively identify usability issues and user behaviours that might not be apparent in controlled environments [46]. Additionally, remote testing can reach a more diverse and representative sample of participants, enhancing the generalizability of the findings [27]. 

### 4.4. Translation Accuracy and Cultural Relevance of PILs

Finally, another issue stands out from the analysis of the user testing process for PILs. According to the European Medicines Agency guidelines, user testing of PILs is typically conducted in the language in which the PIL was originally drafted, usually English. Once this initial user testing is successfully completed, there is no requirement to conduct additional user testing on translated versions of the PIL. This approach presumably aims to facilitate the approval process across different language versions while maintaining consistency in the information provided. However, it also raises concerns about the accuracy and cultural relevance of the translations, as the translated PILs are automatically approved without further testing in each target language. This approach can overlook several critical issues that impact the effectiveness and safety of the PILs [47]. One of the primary challenges is the potential loss of meaning and nuance, as medical terminology, instructions, and warnings may not have direct equivalents in other languages, leading to confusion and misinterpretation. Shashkevich [48] points out that health literacy varies significantly across populations, and translations that do not account for these variations can compromise the effectiveness of health communications. When patients misinterpret dosage instructions or side effects due to poor translation, their safety is at risk [21,49,50]. Cultural relevance and context also play a crucial role in how information is perceived and understood. A PIL that is clear in English may not be as effective in another cultural context. Hall et al. [51] argue that cultural context significantly influences communication and understanding. For example, cultural attitudes towards medication, health beliefs, and common practices can vary widely, affecting how patients interpret the information provided in a PIL. The automatic approval of translated PILs without additional user testing in each target language fails to account for these cultural nuances. This oversight can lead to translated PILs that are technically accurate but linguistically and culturally inappropriate, reducing their effectiveness and potentially causing harm. Furthermore, inconsistencies and mistakes in translation are common when PILs are translated into multiple languages [49,52,53]. Jääskeläinen highlighted that even skilled translators can introduce mistakes or inconsistencies, especially when dealing with complex medical texts [54]. These mistakes can range from minor typographical mistakes to significant mistranslations that alter the meaning of critical information. The automatic approval process assumes that any translation of an initial user-tested version is mistake-free and contextually appropriate, maintaining the same quality. However, without rigorous quality checks, validation, and user testing in each target language, there is a risk that mistakes will go unnoticed, compromising patient safety. One effective method for ensuring translation accuracy is back-translation, a process in which the translated text is retranslated back into the original language by a different translator who was not involved in the initial translation. This approach helps to identify mistakes, ambiguities, and discrepancies, ensuring that the translated text accurately conveys the intended meaning of the original [54]. Finally, user testing should still be required to ensure that the translated materials are comprehensible and effective for the target audience.

While robust translation protocols such as back-translation and review by native speakers can help ensure accuracy and clarity, it is important to acknowledge that implementing these additional steps would inevitably slow down the overall approval process. However, this trade-off is essential for ensuring that the translated materials are not only accurate but also comprehensible and effective for the target audience, thereby enhancing patient safety and understanding. Ultimately, such measures contribute to the long-term goal of improving the quality and usability of PILs across diverse linguistic and cultural contexts [55].

### 4.5. Limitations

A few limitations of this study should be noted. The analysis relies primarily on secondary data from existing literature and legislative documents, which may introduce interpretation bias. Additionally, the study does not fully address the practical challenges, such as the increased time and costs, associated with implementing the suggested improvements in user testing protocols, which may slow down the regulatory process. However, these limitations are mitigated by the comprehensive scope of the review, which draws from a wide range of sources to ensure that key gaps in the regulatory framework are identified. Furthermore, while primary data collection or automated data analysis could enhance future studies, the manual approach allows for a nuanced understanding of complex regulatory texts. Addressing the trade-off between improving user testing procedures and the potential delay in PIL approval is crucial for enhancing patient safety and ensuring that the recommendations made are feasible within existing regulatory constraints.

## 5. Conclusions

Significant progress has been made in regulating PILs, yet critical gaps remain. To further enhance their effectiveness, EU guidelines must prioritize diverse and representative sampling in user testing. Stratified sampling techniques are necessary to ensure that all patient demographics are considered, avoiding the risk of PILs being incomprehensible to certain groups. Additionally, independent third-party testing should be mandated to eliminate potential biases from pharmaceutical companies’ own testing processes, resulting in more reliable and objective evaluations.

Improving translation processes is also vital. Robust protocols, including back-translation and cultural adaptation, will ensure that PILs maintain their clarity and accuracy in different languages and cultural contexts. Furthermore, user testing of translated PILs is essential to verify their usability for non-native speakers and ensure that the translated content is as effective as the original. The EMA must play a more active role in updating and standardizing user testing practices. Current guidelines allow for too much flexibility, leading to inconsistencies across the industry. Regular updates based on the latest research and best practices will be necessary to ensure that PILs are uniformly effective across all member states. In conclusion, strengthening user testing protocols, enhancing translation procedures, and standardizing practices will lead to more accessible and user-friendly PILs. These improvements are critical for promoting patient safety, improving medication adherence, and advancing public health through clearer, more reliable communication in the pharmaceutical industry.

## Figures and Tables

**Figure 1 ijerph-21-01301-f001:**
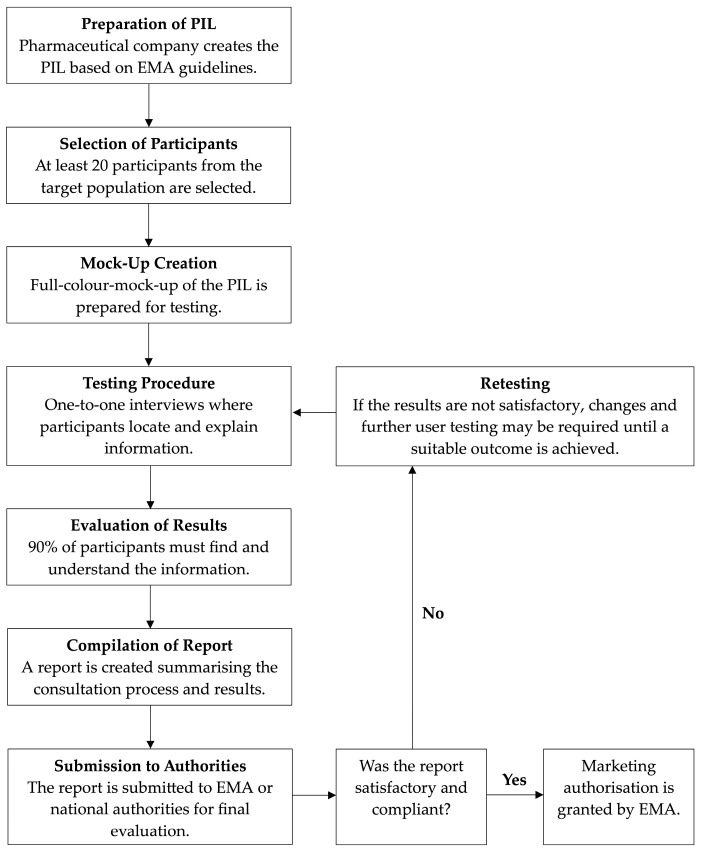
User testing process for PILs in the EU.

## Data Availability

The data utilised in this study are derived from publicly available legislative documents, including Directive 2001/83/EC and its subsequent amendments (Revisions 14 through 14.8), as well as Articles 59(3) and 61(1) of Directive 2001/83/EC as amended by Directive 2004/27/EC. These documents are accessible through the official website of the European Union and relevant legal repositories. No additional datasets were generated or analysed during the current study.

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
