# Peer review of "The Challenges for EU User Testing Policies for Patient Information Leaflets"

_ijerph, 2024, doi:10.3390/ijerph21101301_

Round 1

Reviewer 1 Report

Comments and Suggestions for Authors

I would like to thank the Editor for the invitation to review this manuscript. I believe the topic is of significant interest. However, I would like to offer some suggestions to the author to enhance the quality of the manuscript:

  1. Please ensure that the name, affiliation, and corresponding author information are provided according to the journal's guidelines.
  2. Please format the references according to the journal's requirements.
  3. The introduction is well-written and structured. However, the study's objective should be clearly defined at the end of the introduction.
  4. Dividing the methods section into subsections—such as study design, theoretical framework, study procedures, qualitative data analysis (did you use any specific software?)—could improve readability.
  5. The results section should be labeled as section 3, the discussion as section 4, and the conclusion as section 5.
  6. In the results section, I suggest adding subsections to clarify the findings more effectively. Additionally, a standardized summary table could be introduced to present the qualitative results more clearly.
  7. Lines 228-29: The phrase "When pharmaceutical companies conduct their own user testing, it is akin to asking the fox to guard the henhouse—there is an inherent risk of conflict of interest." should be rephrased to adopt a more neutral tone.
  8. In line with the previous comment, I recommend revising other phrases or sections that could be deemed problematic by third-party companies. While these sections may reflect the reality of the situation, rewording them in a more neutral manner would help avoid potential issues for both the author and the journal.
  9. The note about the "Hawthorne effect" should be included as a sentence within the manuscript, not just as a legend: "The Hawthorne effect refers to the phenomenon where individuals modify their behavior in response to their awareness of being observed, potentially skewing research outcomes (McCarney et al., 2007)." Similarly, the paragraph on back-translation should also be incorporated into the text: "Back-translation is a method used to verify the accuracy and equivalence of translated texts. It involves translating a document back into the original language by a different translator, helping to identify mistakes and ambiguities in the translation (Jääskeläinen, 2016)."
  10. The limitations of the study should be addressed at the end of the discussion section.
  11. The conclusions should be shortened by removing redundant information already presented earlier. Additionally, references should not be included in this section.

Author Response

Dear reviewer,

Thank you for your valuable comments, which I truly feel have greatly improved the article. Your insights were highly appreciated. Please see my replies to your points.

Reviewer comment 1: Please ensure that the name, affiliation, and corresponding author information are provided according to the journal's guidelines.

Author response/action: This was successfully fixed.

Reviewer comment 2: Please format the references according to the journal's requirements.

Author response/action: This was successfully fixed.

Reviewer comment 3: The introduction is well-written and structured. However, the study's objective should be clearly defined at the end of the introduction.

Author response/action: This statement was added to define the aims: "The aim of this study is to critically examine the current EU policies surrounding user testing for PILs and identify gaps that may hinder their effectiveness in ensuring readability and accessibility for diverse patient populations. By evaluating the diversity of user testing samples, the independence of the testing process, and the sufficiency of testing protocols, this research seeks to offer recommendations for improving the robustness of these regulations to enhance patient safety and comprehension".

Reviewer comment 4: Dividing the methods section into subsections—such as study design, theoretical framework, study procedures, qualitative data analysis (did you use any specific software?)—could improve readability.

Author response/action: Methods were divided into sections, specifically: study design, theoretical framework and qualitative data analysis. No software was mentioned as I used manual recording.

Reviewer comment 5: The results section should be labeled as section 3, the discussion as section 4, and the conclusion as section 5.

Author response/action: This has been fixed. 

Reviewer comment 6: In the results section, I suggest adding subsections to clarify the findings more effectively. Additionally, a standardized summary table could be introduced to present the qualitative results more clearly.

Author response/action: I have added sections to the results, specifically:  User Testing Origins and EU Adoption, User Testing Process and Criteria, Reporting and Implementation Challenges. I have also added a flowchart summarising the various steps of the user testing process (See Microsoft Word file attached).

Reviewer comment 7: Lines 228-29: The phrase "When pharmaceutical companies conduct their own user testing, it is akin to asking the fox to guard the henhouse—there is an inherent risk of conflict of interest." should be rephrased to adopt a more neutral tone.

Author response/action: I have changed this into: "When pharmaceutical companies conduct their own user testing, there is a potential for conflict of interest, which may influence the objectivity of the process".

Reviewer comment 8: In line with the previous comment, I recommend revising other phrases or sections that could be deemed problematic by third-party companies. While these sections may reflect the reality of the situation, rewording them in a more neutral manner would help avoid potential issues for both the author and the journal.

Author response/action: I have applied your suggestion to give those statements a neutral tone. 

Reviewer comment 9: The note about the "Hawthorne effect" should be included as a sentence within the manuscript, not just as a legend: "The Hawthorne effect refers to the phenomenon where individuals modify their behavior in response to their awareness of being observed, potentially skewing research outcomes (McCarney et al., 2007)." Similarly, the paragraph on back-translation should also be incorporated into the text: "Back-translation is a method used to verify the accuracy and equivalence of translated texts. It involves translating a document back into the original language by a different translator, helping to identify mistakes and ambiguities in the translation (Jääskeläinen, 2016)."

Author response/action: They are now integrated into the text.

Reviewer comment 10: The limitations of the study should be addressed at the end of the discussion section.

Author response/action: Limitations were addressed in an appropriate section as suggested: "A few limitations of this study should be noted. The analysis relies primarily on secondary data from existing literature and legislative documents, which may introduce interpretation bias. Additionally, the study does not fully address the practical challenges, such as the increased time and costs, associated with implementing the suggested improvements in user testing protocols, which may slow down the regulatory process. However, these limitations are mitigated by the comprehensive scope of the review, which draws from a wide range of sources to ensure that key gaps in the regulatory framework are identified. Furthermore, while primary data collection or automated data analysis could enhance future studies, the manual approach allows for a nuanced understanding of complex regulatory texts. Addressing the trade-off between improving user testing procedures and the potential delay in PIL approval is crucial for enhancing patient safety and ensuring that the recommendations made are feasible within existing regulatory constraints".

Reviewer comment 11: The conclusions should be shortened by removing redundant information already presented earlier. Additionally, references should not be included in this section.

Author response/action: The conclusions have been shortened a great deal removing information presented earlier. References in this part were taken off.

Thank you very much once again for all the precious help!

Reviewer 2 Report

Comments and Suggestions for Authors

This article comments on the EU procedure for user testing of PILs, pointing out some shortcomings. It does not mention examples of where these shortcomings have led to problems but references are made to other literature where such problems are documented.

While the recommended changes in practice seem justified, they will not always be easy to implement. For example, user testing of every translation would slow down the procedure tremendously and maybe the text should acknowledge this challenge. 

A source that hasn't been mentioned and that remains interesting, even though it is dated, is Ahlfeldt et al. 2006 (Literature Review on Patient-Friendly Documentation Systems (uni-bielefeld.de)).

PILs are called package leaflets in the EU. It would be good to mention this early on in the text, for the EU term is later used in lines 114 and 158.

Lines 73-77: While the centralised procedure has by and large become the standard, you may mention here that the alternative procedures (decentralised, mutual recognition) still exist, the more so because you mention them in lines 195-196.

Some details:

- abstract, line 8: "The European Medicines Agency (EMA) oversees etc." > start the sentence with: "In the European Union, the European Medicines Agency etc."

- Discussion: to make the separate arguments stand out, it would be good to structure this part in separate paragraphs.

Comments on the Quality of English Language

It is true that user testing organized by the company that markets the medicinal product can easily lead to bias. The wording here could be more neutral, however ('a spectre that looms large', 'the fox that guards the henhouse' are expressions reminiscent of journalistic prose).

Details:  22: Eu > EU  / 107: PILS > PILs /  254 add comma before which / 313 highlighting > highlighted

Author Response

Dear Reviewer,

Thank you for your valuable comments, which I feel have greatly improved the article. Your insights were highly appreciated. Please find attached a point-to-point response.

Reviewer's comment 1: "It does not mention examples of where these shortcomings have led to problems but references are made to other literature where such problems are documented".

Author's response: "Some examples have now been included in the discussion to exemplify which problems were connected to this issue. E.g. "Moreover, previous literature has demonstrated that non-representative user samples can lead to significant usability issues, especially in patient information leaflets. For example, studies by Albassam and Hughes as well as Liu et al. have identified gaps in PIL comprehension related to literacy levels and cultural background, where non-diverse samples led to poor real-world usability, ultimately compromising patient safety.

Reviewer's comment 2: "While the recommended changes in practice seem justified, they will not always be easy to implement. For example, user testing of every translation would slow down the procedure tremendously and maybe the text should acknowledge this challenge. ".

Author's response: I agree with this. Thank you. This has been added now: "...it is important to acknowledge that implementing these additional steps would inevitably slow down the overall approval process. However, this trade-off is essential for ensuring that the translated materials are not only accurate but also comprehensible and effective for the target audience, thereby enhancing patient safety and understanding. Ultimately, such measures contribute to the long-term goal of improving the quality and usability of PILs across diverse linguistic and cultural contexts".

Reviewer's comment 3: A source that hasn't been mentioned and that remains interesting, even though it is dated, is Ahlfeldt et al. 2006 (Literature Review on Patient-Friendly Documentation Systems (uni-bielefeld.de)).

Author's response: Thank you. This was a very helpful and interesting source which I feel enriches the study. It was added to the literature review.

Reviewer's comment 4: PILs are called package leaflets in the EU. It would be good to mention this early on in the text, for the EU term is later used in lines 114 and 158.

Author's response: This has been fixed.

Reviewer's comment 5: Lines 73-77: While the centralised procedure has by and large become the standard, you may mention here that the alternative procedures (decentralised, mutual recognition) still exist, the more so because you mention them in lines 195-196.

Author's response: I have added an explanation earlier in the text now. "In the European Union, pharmaceutical companies can pursue different procedures for obtaining marketing authorisation. The centralised procedure, overseen by the EMA, allows companies to submit a single application for EU-wide approval, mandatory for certain medicines, including biotechnology products and orphan drugs. Alternatively, the decentralised procedure enables companies to apply for authorisation in multiple member states simultaneously, with one reference member state (RMS) leading the as-sessment, and other concerned member states (CMS) either accepting or rejecting the decision. This route is typically used for medicines not previously authorised in any EU country and outside the centralised procedure’s scope. The mutual recognition procedure, by contrast, is applied when a product has already been authorised in one EU member state, allowing the company to extend the marketing authorisation to other member states, which then “mutually recognise” the existing authorisation. Each option offers a tailored approach depending on the product type and market strategy [15]. In whichever route is chosen to apply for authorisation, whether through the centralised, decentralised, or mutual recognition procedure, the EMA ultimately conducts a scientific evaluation and provides its opinion to the European Commission for the final decision".

Reviewer's comment 6: 

- abstract, line 8: "The European Medicines Agency (EMA) oversees etc." > start the sentence with: "In the European Union, the European Medicines Agency etc."

- Discussion: to make the separate arguments stand out, it would be good to structure this part in separate paragraphs. It is true that user testing organized by the company that markets the medicinal product can easily lead to bias. The wording here could be more neutral, however ('a spectre that looms large', 'the fox that guards the henhouse' are expressions reminiscent of journalistic prose). Details:  22: Eu > EU  / 107: PILS > PILs /  254 add comma before which / 313 highlighting > highlighted

Author's response: All these comments were fixed as suggested. Also, the discussion is now structured into separate sections: 

4.1 Participant Selection and Diversity in User Testing, 4.2 Potential Biases in the User Testing Process, 4.3 The Controlled Environment in User Testing, 4.4 Translation and Cultural Relevance of PILs, 4.5 Limitations

Once again, thank you very much for all your precious help!

Round 2

Reviewer 1 Report

Comments and Suggestions for Authors

The author provided appropriate edits to the manuscript and implemented quality. The manuscript is ready for publication